

# Partitioning of ionic surfactants in aerosol droplets containing glutaric acid, sodium chloride, or sea salts

Alison Bain[1,2*], Kunal Ghosh[3], Konstantin Tumashevich[3], Nønne L. Prisle[3], and Bryan R. Bzdek[1*]

[1]School of Chemistry, University of Bristol, Bristol, BS8 1TS, United Kingdom
[2]Department of Chemistry, Oregon State University, Corvallis, 97331, United States
[3] Center for Atmospheric Research, University of Oulu, Oulu, P.O. Box 4500, 90014, Finland

*Correspondence to*: Alison Bain (alison.bain@oregonstate.edu) and Bryan R. Bzdek (b.bzdek@bristol.ac.uk)

**Abstract.** Sea spray aerosol is the largest contributor to atmospheric aerosol by mass and contains mixtures of inorganic salts and organics. The chemically complex organic fraction can contain soluble and surface-active organics, and field studies commonly identify ionic surfactants in aerosol samples. In macroscopic solutions, divalent cations present in sea spray have been found to alter the partitioning of ionic surfactants. Furthermore, the high surface area-to-volume (SA-V) ratio of aerosol droplets may lead to depletion of surfactant from the bulk, requiring more surfactant, relative to its volume, to lower the surface tension of a droplet compared to a macroscopic solution. Here, we investigate the partitioning of model ionic surfactants (sodium dodecylsulfate, an anionic surfactant, and cert tetrammonium bromide, a cationic surfactant) in 6 – 10 μm radius droplets containing glutaric acid, NaCl, or sea spray mimic cosolutes. Surface tension measurements are compared to two independent partitioning models which account for the SA-V ratio of the droplets. Salting out of the ionic surfactants leads to strong bulk depletion in 6 – 10 μm radius droplets. No difference in droplet surface tension was observed between NaCl and sea spray mimic cosolutes. The total concentration of ionic surfactant required to reach the minimum surface tension in these droplets (water activity ~0.99) was 2.0 ± 0.5 mM when the macroscopic critical micelle concentration is < 2 mM. These results are consistent with previous observations in droplets containing nonionic surfactants. The partitioning of ionic surfactants in salt-containing droplets has implications for cloud droplet activation and chemistry occurring at the interface of sea spray aerosol.

## 1 Introduction

Aerosol in the atmosphere can scatter and absorb radiation, directly affecting climate, as well as alter cloud microphysics by serving as cloud condensation nuclei (CCN), indirectly impacting climate. Sea spray is a major source of surfactants in atmospheric aerosol (De Leeuw et al., 2011). Ambient aqueous aerosol contains a complex mixture of organic and inorganic species. For example, the sea surface microlayer contains salts as well as a high concentration of organic molecules. Bubble bursting and jet drop aerosol generation pathways have been shown to produce droplets with enriched in surfactant compared to the sea surface microlayer (Bertram et al., 2018; Burdette et al., 2022; Cochran et al., 2016; Frossard et al., 2019; Wang et al., 2017). Surfactants are known to reduce surface tension at macroscopic aqueous interfaces and may also lower the surface tension of microscopic droplets if present in sufficiently high quantities (Bain et al., 2023b; Bzdek et al.,





2020; Jacobs et al., 2024). Field studies often find ionic surfactants make up more than half of the total surfactant fraction (Gérard et al., 2016, 2019). Sea water is also a source of divalent cations like calcium. Bridging interactions (i.e., the tethering of organics to divalent cations) between $Ca^{2+}$ and anionic surfactants leads to co-adsorption at interfaces, and can result in an excess of both $Ca^{2+}$ and surface-active material at interfaces (Carter-Fenk et al., 2021; Hasenecz et al., 2019; Jayarathne et al., 2016).

For an aerosol droplet to become activated into a cloud droplet, it must grow to a critical size, which requires a critical supersaturation in atmospheric relative humidity. The Köhler equation is commonly used in atmospheric science to predict this activation barrier for droplets of a known size and composition. The Köhler equation consists of two terms: the Raoult term, which accounts for the solute effect, and the Kelvin term, which accounts for the surface curvature (Seinfeld et al., 2016). The Kelvin term includes the surface tension of the droplet, which is often assumed equal to the value for pure water during activation (Tao et al., 2012). This assumption is historically considered reasonable because near the point of cloud droplet activation the droplet becomes very dilute (Sorjamaa et al., 2004), but is at odds with field measurements which find highly surface-active material that can lower the surface tension of macroscopic solutions with millimolar range concentrations in collected aerosol samples (Burdette and Frossard, 2021; Gérard et al., 2016, 2019). Furthermore, a growing number of field studies comparing the number of predicted CCN using the Köhler equation to the number of measured CCN have found that lowering the surface tension to $40 - 60$ mN m$^{-1}$ (thereby lowering the barrier to cloud droplet formation) results in better agreement between predictions and measurements (Fan et al., 2024; Good et al., 2010; Irwin et al., 2010; Ovadnevaite et al., 2017). Accurate representations of aerosol surface tension during hygroscopic growth are necessary as shortwave cloud radiative forcing predictions are sensitive to descriptions of aerosol surface tension (Prisle et al., 2012).

Salts greatly affect the surface partitioning of ionic surfactants in macroscopic solutions. For example, the partitioning kinetics of ionic surfactants sodium dodecyl sulfate (SDS) and cert tetrammonium bromide (CTAB) are impacted by the concentration of NaCl in solution (Nozière et al., 2014; Qazi et al., 2020; Rohde and Sackmann, 1979; Weinheimer et al., 1980). Moreover, the addition of NaCl to SDS or CTAB solutions also greatly impacts the critical micelle concentration (CMC) and enhances the equilibrium and surface concentration, commonly referred to as salting out (Prosser and Franses, 2001; Qazi et al., 2020). Additionally, divalent cations such as $Ca^{2+}$ are known to alter the surface properties of anionic surfactants in solution (Cross and Jayson, 1994; Penfold and Thomas, 2022).

Predicting the surface tension of macroscopic aqueous mixtures of organics and inorganics is non-trivial and cannot be accomplished with simple mass or volume fraction mixing rules (Boyer et al., 2017; Kleinheins et al., 2024; May et al., 2018; Tuckermann, 2007; Wu et al., 2019). Compared to macroscopic solutions, predicting the surface tension of aerosol becomes even more challenging. The surface-active nature of organic molecules, in addition to the small size, and therefore large surface area-to-volume (SA-V) ratio, of aerosol droplets means that surfactants can become depleted in the droplet bulk (Bain et al., 2023b; Bzdek et al., 2020; Jacobs et al., 2024; Prisle, 2021). This depletion has been experimentally observed for droplets in air containing nonionic surfactants spanning a wide range of chemical structures and surfactant properties. Models predicting surfactant partitioning in droplets have also been tested against droplet measurements for these nonionic




surfactants (Bain et al., 2023b, 2024a; Bzdek et al., 2020). However, both nonionic and ionic surfactants have been characterized and quantified in ambient aerosol samples (Frossard et al., 2019; Gérard et al., 2016, 2019).

In this work, we investigate the surface tension of aqueous aerosol droplets containing atmospheric proxy anionic or cationic surfactants mixed with glutaric acid, NaCl, or a sea spray mimic. We measure the surface tensions of picolitre volume droplets suspended in air as well as of macroscopic solutions, systematically altering the surfactant composition, and compare these results to predictions from two independent partitioning models. This systematic, bottom-up approach provides a framework to understand the partitioning of ionic surfactants in realistic aerosol droplets, which contain complex mixtures of surfactants and cosolutes.

## 2 Methods

### 2.1 Sample preparation

Solutions containing cosolutes and one surfactant were prepared for macroscopic and droplet surface tension measurements. NaCl (Sigma-Aldrich BioXtra, >99.5%) and glutaric acid (Sigma-Aldrich 99%) cosolutes were used without further purification. Sea spray mimic (herein referred to as sea spray) was made up from NaCl (Sigma-Aldrich BioXtra, >99.5%), $MgCl_2$ ($MgCl_2 \cdot 6H_2O$, Biosciences), $CaCl_2$ ($CaCl_2 \cdot 2H_2O$, Argos Organics, 99+% ACS reagent), $Na_2SO_4$ (Fisher Scientific), and $MgSO_4$ (Fisher Scientific). Sea spray was made to match the inorganic ion ratios using the five most abundant ions of the fine fraction sea spray aerosol as reported in the literature (Jayarathne et al., 2016) (ion mole fractions of 0.493 $Cl^-$, 0.410 $Na^+$, 0.057 $Mg^{2+}$, 0.012 $Ca^{2+}$ and 0.028 $SO_4^{2-}$). The cationic surfactant CTAB (BioXtra >99% Sigma Aldrich) was used without further purification, but the anionic surfactant SDS (ultra-pure MP Biomedicals, LLC) was purified by three consecutive recrystallizations before use. After recrystallization, surface tension measurements of SDS no longer showed a dip in surface tension around the CMC, indicating the surface-active impurity had been removed. Solutions for macroscopic and droplet measurements contained one surfactant (SDS or CTAB) at a range of concentrations and one cosolute (0.9 M glutaric acid, 0.5 M NaCl, or 0.48 M sea spray). The sea spray concentration was chosen to be 0.48 M total salt so that the molar amounts of anions and cations would be equal to that of 0.5 M NaCl. These concentrations were chosen to represent water activity near saturation ($a_w$=0.99 for each cosolute). All solutions were prepared with deionized water.

### 2.2 Droplet surface tension measurements

The surface tension of single droplets was determined by the coalescence method using holographic optical tweezers, which has been previously described (Bain et al., 2023b, a; Bzdek et al., 2016, 2020). The surface tension of the droplet ($\sigma$, Eq. 1) is related to the oscillation frequency ($\omega_l$) of the surface mode order $l$, which is excited upon droplet coalescence, as well as the droplet's radius ($r$) and density ($\rho$). The radius and refractive index are determined by fitting the cavity-enhanced Raman spectrum of the composite droplet (Preston and Reid, 2013, 2015). The droplet's density and viscosity were then determined using parameterizations.



$$\sigma = \frac{r^3 \rho \omega_l{}^2}{l(l-1)(l+2)} \tag{1}$$

The observed oscillation frequency ($\omega_l^*$) is corrected for viscous damping ($\eta$), assuming the viscosity of the surfactant-containing droplet is equal to that of a droplet with the same concentration of primary solute (NaCl, glutaric acid, or sea spray).

$$\omega_l^* = \sqrt{\omega_l^2 - \tau_l^{-2}} \tag{2}$$

$$\tau_l = \frac{r^2 \rho}{(l-1)(2l+1)\eta} \tag{3}$$

Parameterizations for the density, viscosity, and composition of glutaric acid and NaCl droplets were taken from the literature (Bzdek et al., 2016; Rumble, 2021; Song et al., 2016). Parameterizations for sea spray (Table S1 and Figs. S1) were determined by fitting density (density meter, Densito METTLER TOLEDO) and refractive index (n(589 nm), Palm Abbe digital refractometer (PA201, MISCO)) measurements of the sea spray mimic. The viscosity of sea spray is assumed to be equal to the viscosity of NaCl for the same total molar concentration. This is a reasonable assumption since the solute is dilute ($a_w$=0.99) in all droplets. The concentration of surfactant in the droplet was then determined using the molar ratio of surfactant to cosolute in the nebulized solution (Bain et al., 2023b; Bzdek et al., 2020).

### 2.3 Macroscopic surface tension measurements

The equilibrium surface tension of macroscopic solutions containing surfactant and cosolute were measured with the Wilhelmy plate method (Krüss, K100) at 25 ± 1 °C. Reported surface tensions are an average of three repeat measurements. The macroscopic data were fit in the region of decreasing surface tension with increasing surfactant concentration to the Langmuir isotherm and equation of state (Eq. 4).

$$\sigma = \sigma_0 + nRT\,\Gamma_{max} \ln\left(1 - \frac{\Gamma}{\Gamma_{max}}\right) = \sigma_0 + nRT\,\Gamma_{max} \ln\left(1 - \frac{K_{eq}c}{1 + K_{eq}c}\right) \tag{4}$$

In the Langmuir isotherm, $\Gamma$ is the equilibrium surface excess at a specific surfactant concentration, $\Gamma_{max}$ is the maximum surface excess, $K_{eq}$ is the equilibrium partitioning constant, $c$ is the surfactant concentration, $\sigma_0$ is the surface tension without surfactant present, $R$ is the gas constant, $T$ is the temperature, and $n$ is the van't Hoff factor for the surfactant at the surface. $n$ is typically set to two for ionic surfactants in water, but one in the presence of excess electrolyte. Here, $n = 2$ for surfactant mixtures with glutaric acid and $n = 1$ for surfactant mixtures with NaCl or sea spray. The surface tension without surfactant





present is set to 63.0 mN m$^{-1}$ for ternary mixtures with 0.9 M glutaric acid and to 72.0 mN m$^{-1}$ for ternary mixtures with 0.5
M NaCl or sea spray. The partitioning models are not sensitive to small differences in the choice of $\sigma_0$. Macroscopic
measurements in the region of the surface tension plateau (concentration > CMC) were fit with a straight line. The CMC is
then taken as the intersection of this straight line with the Langmuir isotherm and the surface tension at surfactant
concentrations greater than the CMC where the minimum surface tension has been reached ($\sigma^{min}$), is taken to be the average
of all the surface tensions in this region.

## 2.4 Droplet bulk depletion

### 2.4.1 Simple Kinetic Partitioning Framework

The bulk surfactant concentration in picolitre and smaller volume droplets can become depleted due to the high SA-V ratio.
The Simple Kinetic model, first developed by Alvarez et al., uses a mass balance for surfactant partitioned to the interface
and dissolved in the bulk in combination with an isotherm model to express the depleted bulk concentration at equilibrium
(Eq. 5) (Alvarez et al., 2012; Bain et al., 2024a).

$$\frac{C_{eff}}{C_i} = \frac{1}{2}\left(1 - \zeta - \frac{\zeta}{f}\right) + \frac{1}{2}\sqrt{\left(1 - \zeta - \frac{\zeta}{f}\right)^2 + 4\frac{\zeta}{f}} \tag{5}$$

The depleted bulk concentration, or effective concentration ($C_{eff}$), is normalized by the initial bulk concentration
($C_i$, i.e., total surfactant concentration) and is written as a function of two dimensionless parameters, $f$ and $\zeta$. For the case of
a spherical droplet with surfactant dissolved in the interior (Alvarez et al., 2012),

$$f = \frac{3K_{eq}\Gamma_{max}}{r} \tag{6}$$

$$\zeta = \frac{3\Gamma_{max}}{C_i r} \tag{7}$$

where $r$ is the radius of the droplet. The parameters $\Gamma_{max}$ and $K_{eq}$ can be found by fitting macroscopic surface tension
measurements to the Langmuir isotherm (Eq. 4) (Eastoe and Dalton, 2000).

To predict the surface tension in a droplet with depleted surfactant concentration due to bulk-to-surface partitioning,
$c$ in Eq. 4 is replaced with $C_{eff}$. Note, as droplet radius increases and $C_{eff}$ approaches $C_i$, the predicted surface tension
approaches the macroscopic surface tension from the Langmuir isotherm fit. When the predicted surface tension becomes
equal to the average surface tension measured for macroscopic solutions at concentrations greater than the CMC, the surface



tension is set equal to this average value for all larger surfactant concentrations since the droplet size is not expected to alter

the surface tension after the droplet bulk CMC is reached for the picolitre volume droplets studied here.

### 2.4.2 Monolayer Partitioning Framework

The Monolayer Model developed by Malila and Prisle calculates the surface tension of aqueous droplets based on the

composition of the bulk phase, which is determined from the total droplet composition by accounting for size-dependent

bulk-to-surface partitioning (Malila and Prisle, 2018). In the Monolayer Model, a finite-sized spherical droplet with radius $r$

is comprised of a surface monolayer with thickness $\delta$ and an interior (bulk) of radius $r - \delta$. The surface is described as a

separate liquid phase with a composition distinct from that of the bulk.

The compositions of the droplet bulk (superscript $b$) $\boldsymbol{\chi}^b = (\chi_1^b, \chi_2^b, \dots)$ and surface (superscript $s$) $\boldsymbol{\chi}^s = (\chi_1^s, \chi_2^s, \dots)$

are calculated iteratively using the semi-empirical relation

$$\sigma(\boldsymbol{\chi}^b) = \frac{\sum_i \chi_i^s v_i \sigma_i}{\sum_i \chi_i^s v_i} \tag{8}$$


between the droplet surface tension $\sigma$, parameterized in terms of the composition of the bulk (left side Eq. 8), and weighted

by the volumes of individual components in the surface (right side Eq. 8). Here, $\chi_i^b$ and $\chi_i^s$ are the bulk and surface mole

fractions, corresponding to molar amounts $n_i^b$ and $n_i^s$, respectively, $v_i$ are the molecular volumes, and $\sigma_i$ are pure compound

surface tensions, of each droplet component $i$. Details of model assumptions and boundary conditions have been described

previously (Bain et al., 2023b; Bzdek et al., 2020; Malila and Prisle, 2018).

Here, Eq. 4 was used to parameterize surface tension for the left-hand side of Eq. 8. We simplify Eq. 4 by setting

$b = nRT \, \Gamma_{max}$. Surface tensions when the surfactant concentration is zero ($\sigma_0$) were again set to 63.0 mN m$^{-1}$ for ternary

mixtures with glutaric acid and 72.0 mN m$^{-1}$ for ternary mixtures with NaCl and sea spray mimic. Pure component physical

parameters required for all components of the droplet are provided in Table S2. In the case of sea spray, the E-AIM (Dutcher

et al., 2010) predictions of the surface tension using the ion mole ratios and the measured densities are extrapolated to the

pure component. We treat sea spray as a single cosolute, using its mole averaged molecular mass. The surface tensions of

pure, non-aqueous surfactants are not known and are approximated by the surface tensions at the CMC of the surfactant in a

binary aqueous solution, $\sigma_{CMC}$. This assumption corresponds to assuming that a complete, pure monolayer with

$\chi_{surfactant}^s = 1$ has formed at the CMC and may therefore in some cases lead to discontinuous changes in droplet surface

tension σ and $\chi_i^s$ when $\chi_{surfactant}^b$ reaches the CMC.

Figure S2 shows the macroscopic experimental datasets fit with the Langmuir isotherm. Monolayer Model

predictions for large radius droplets (100 μm and 100 cm) are also overlayed to show that as droplet radius is increased, the

Monolayer Model predictions tend towards the original parameterization.



**2.5 Droplet bulk depletion**

The agreement between the experimentally determined surface tensions and model predictions are quantified with the mean absolute error (MAE), defined as:

$$MAE = \frac{1}{N}\sum_{i=1}^{n}|M_i - E_i| \tag{9}$$

where $M_i$ and $E_i$ are the model prediction and experimental surface tension, respectively, for data point i, and $N$ is the total number of datapoints. If $M_i^{10} < E_i < M_i^6$ (where $M_i^6$ and $M_i^{10}$ are the model predictions for 6 and 10 µm radii, respectively), the residual is set to zero. If $E_i > M_i^6$ the residual is $M_i^6 - E_i$; if $E_i < M_i^{10}$ the residual is $M_i^{10} - E_i$. The MAEs are

calculated after removing datapoints where the concentration is greater than the average of the effective droplet CMCs (i.e. the concentration where the surface tension plateaus). When the concentration is greater than the effective droplet CMC, nonequilibrium surface concentrations impact the measured surface tension.

The mean absolute scaled error (MASE) is used to compare the models' abilities to predict the experimental data against one another:

$$MASE = \frac{MAE_{Monolayer}}{MAE_{Simple\ Kinetic}}. \tag{10}$$

**3 Results and Discussion**

In this work, CTAB and SDS were chosen as commercially available cationic and anionic representatives of atmospheric surfactants. SDS is particularly environmentally relevant due to its widespread use in soaps and cleaning agents and later release during water treatment (Cochran et al., 2016; Radke, 2005). To date, cationic surfactants found in aerosol and sea surface microlayer samples during field campaigns have been quantified, but their chemical structures have not yet been

identified (Burdette et al., 2022; Burdette and Frossard, 2021; Gérard et al., 2016, 2019). Generally, oxygen to carbon ratios measured with mass spectrometry show the presence of aliphatic surfactants as well as lignin-like and carboxyl-rich alicyclic molecules (Burdette et al., 2022). Glutaric acid and NaCl were chosen as cosolutes. Glutaric acid represents a soluble organic molecule that is often used in laboratory experiments as a proxy for oxidized organic material, and it has also been identified in ambient aerosol (Bondy et al., 2018; Wu et al., 2019). NaCl is the most abundant salt in sea water and therefore

sea spray aerosol (Gong et al., 2002; Jayarathne et al., 2016). Since sea spray contains a mixture of ions, including $Ca^{2+}$ and $Mg^{2+}$, a sea spray mimic was used as an additional cosolute to determine if the concentrations of divalent cations in sea spray aerosol near the point of cloud droplet activation would alter the partitioning of surfactants in droplets. Sea spray was made up to have the inorganic ion composition of fine mode sea spray droplets, which have the greatest enhancement of divalent cation concentration relative to sea water (Jayarathne et al., 2016).



Figure S3 shows macroscopic surface tension data for SDS and CTAB each with the three studied cosolutes. Macroscopic measurements of aqueous SDS and CTAB without a cosolute agree well with literature results (Zdziennicka et al., 2012). The NaCl and sea spray mimic cosolutes clearly influence partitioning of SDS and CTAB, resulting in CMCs and Langmuir isotherm parameters (Table 1) that differ by orders of magnitude from the parameters associated with the respective binary aqueous solution, as well as with glutaric acid cosolute solutions. The properties of ionic surfactants can be

substantially altered by salts (Eastoe et al., 2000; Iyota and Krastev, 2009; Kumar et al., 2012; Prosser and Franses, 2001; Qazi et al., 2020). Solutions containing the counter ion of the ionic surfactant reduce the surfactant's solubility, lowering the CMC. Anionic surfactant properties are also affected by divalent cations (Eastoe et al., 2000; Eastoe and Dalton, 2000; Penfold and Thomas, 2022). The effect of cosolute on the surfactant parameters is larger here for ionic surfactants than previously observed with nonionic surfactants. For nonionic surfactants, glutaric acid increased the CMC in macroscopic

solutions (generally within a factor of 4) while NaCl slightly reduced it (within a factor of 2.5) (Bain et al., 2023b). In contrast, for the ionic surfactants SDS and CTAB, the addition of 0.9 M glutaric acid reduces the CMC (by approximately 30% for SDS and 50% for CTAB), whereas in solutions containing 0.5 M NaCl or 0.5 M cations/anions, the macroscopic CMC is reduced by more than an order of magnitude compared to the binary aqueous surfactant case.

    Experimentally determined surface tensions for 6 – 10 µm radius droplets containing either SDS or CTAB and one

of three cosolutes are shown in Fig. 1 along with droplet surface tension predictions from the Simple Kinetic Model (Alvarez et al., 2012) and the Monolayer Model (Malila and Prisle, 2018). The predicted surface tensions shown in Fig. 1 are for droplets at 6 µm (solid) and 10 µm (dashed) radius, with the shaded region between these two predictions encompassing the experimental droplet measurement range. Vertical lines indicate the CMC determined from macroscopic measurements. The uncertainty on the droplet surface tension measurements, propagated from the uncertainty on the droplet radius and

oscillation frequency, is less than 0.5 mN m$^{-1}$.

    When bulk surfactant depletion occurs, more surfactant is required to reach the CMC in the droplet bulk because a substantial fraction of molecules is removed from the bulk to populate the interface. Here we use the term "effective CMC" to describe the total surfactant concentration required to reach a plateau in droplet surface tension measurements. For SDS with glutaric acid cosolute (Fig. 1A), the macroscopic CMC and effective CMC are in close agreement. This agreement

indicates that there is little bulk depletion of SDS in these droplets, consistent with previous observations for surfactants with CMCs above about 5 mM (Bain et al., 2023b). In Fig. 1 B – F, larger differences between the macroscopic CMCs and the effective CMC are observed, with the droplet measurements for these systems exhibiting a mean effective CMC of 2.0 ± 0.5 mM (Table 2). These effective CMCs are consistent with previous measurements for effective CMC in 6 – 9 µm radius droplets containing strong nonionic surfactants (effective CMC = 2.0 ± 0.3 mM when macroscopic CMC < 5 mM ) (Bain et

al., 2023b, 2024a). Figure S4 shows the predicted magnitude of depletion for the ionic surfactant-cosolute systems over the experimental radii and total surfactant concentrations investigated for each surfactant system. Only the SDS-glutaric acid system shows minimal depletion over the investigated radius and concentration range.



Model predictions for 6 – 10 μm radius droplets from both the Monolayer and Simple Kinetic Models in Fig. 1 reach the minimum surface tension at approximately the same total surfactant concentration as the estimated apparent CMC from the experimental droplet data (Table 2). Figure 2 compares the effective droplet CMCs predicted by the two models to those observed in the droplet measurements. The effective CMC predicted by the Simple Kinetic Model is generally closer to the experimentally determined effective CMC, but both models agree with the experimental data within a factor of two.

Although both models predict bulk depletion and similar total surfactant concentrations to reach the minimum surface tension for each surfactant system in Fig. 1, the surfactant concentration-dependent trends in surface tension for each model are different. Generally, at low surfactant concentrations, before the effective CMC, the Monolayer Model predicts the reduction of surface tension with increased concentration to begin at lower concentrations and occur more gradually than the droplet measurements and the Simple Kinetic Model predictions. For SDS and glutaric acid (Fig. 1 panel A), the Monolayer Model slightly overpredicts the surface tension (by about 5 mN m$^{-1}$). Here, the droplet data appear to be biased low as the droplet measurements are also slightly lower than the macroscopic measurements. This bias may be due to an enrichment of surfactant at the droplet interface after coalescence of the two trapped droplets, transiently lowering the surface tension in our observation window (~100 μs after coalescence). The Simple Kinetic Model also slightly overpredicts the surface tension in the SDS and glutaric acid mixtures, but in panels B – F the Simple Kinetic Model is in good agreement with the experimental data below the effective CMC. The mean absolute errors (MAEs) for the Simple Kinetic Model (Table 3) are within 3.5 mN m$^{-1}$. Interestingly, if the macroscopic surface tension minimum limit is not imposed, the surface tension prediction from the Simple Kinetic Model continues to capture the trend in the experimental data (Fig. S5).

Surface tension measurements in 6 – 10 μm radius droplets, as well as Monolayer and Simple Kinetic Model predictions show little difference when NaCl cosolute is replaced with sea spray. Trends in experimentally determined surface tensions and model predictions are similar for NaCl and sea spray cosolutes for each surfactant. For SDS, the Monolayer and Simple Kinetic Models predict differences in effective CMCs between NaCl and sea spray cosolutes of 0.09 and 0.68 mM, respectively. For CTAB, the Monolayer Model and experimental data agree exactly for the effective CMC and the difference in effective CMC is 0.24 mM for the Simple Kinetic Model and experimental data. The effective CMCs determined experimentally for NaCl and sea spray cosolutes are separated by 0.7 mM for surfactant SDS and are identical for CTAB. The low divalent cation concentration in the sea spray droplets at $a_w$=0.99 does not significantly impact the surfactant partitioning in these droplets, indicating that NaCl cosolute can be used to approximate the salt component of sea spray aerosol to understand the partitioning of surfactants in aerosol containing ionic surfactants near the point of cloud droplet activation.

When the total surfactant concentration is sufficient to lower the droplet surface tension to the minimum value, large discrepancies are observed between the models (which are limited by the minimum surface tension of the macroscopic solutions) and the experimental measurements. Under high surface concentrations droplet coalescence is expected to form a compressed film at the droplet surface (Bain et al., 2023b; Bzdek et al., 2020). Figure 3 shows the difference in minimum surface tension ($\Delta\sigma^{min} = \sigma_{bulk}^{min} - \sigma_{droplet}^{min}$) as a function of macroscopic CMC. For nonionic surfactants, a clear trend in





$\Delta\sigma^{min}$ with CMC was previously observed (Bain et al., 2023b). The CTAB systems investigated here follow this trend closely. SDS systems also show a trend in increased $\Delta\sigma^{min}$ with increasing CMC. However, the absolute values are offset higher compared to the rest of the surfactants. The affinity for the surface appears to play a role in the divergence observed in the minimum surface tension in droplets using the droplet coalescence method. However, further investigation is required to understand the offset for droplets containing SDS.

MAE for the agreements of the models with the experimentally measured droplet surface tensions and MASE were calculated to compare the two models' abilities to predict the experimental data. MAEs and MASEs are shown in Table 3. Generally, the Simple Kinetic Model has lower MAEs than the Monolayer Model resulting in MASEs between 1.33 and 6.50. SDS with glutaric acid as well as CTAB with glutaric acid and sea spray had MASE < 2 indicating that the models agree or disagree with the experimental data about equally as well as one another. The remaining systems have MASEs > 2 indicating that the Simple Kinetic Model agrees with the experimental data more than twice as well as the Monolayer Model. The underprediction of surface tension for micron sized droplets containing ionic surfactants and salt cosolutes is likely due to an incomplete description of salting out. The Monolayer Model assumes that both the ionic surfactant and its counter ion partition to the interface together, which may not be the case under high ionic strength conditions. The Monolayer Model also uses a spherical approximation to determine the volume of the surfactant monomers, which may overestimate the amount of space each monomer occupies at the interface, resulting in an overestimate of surface coverage and thus an underprediction of surface tension. Additionally, the Monolayer Model uses the subcooled density of pure surfactants as an input parameter, which may not well describe the density of the surfactants at the droplet interface. Although the Langmuir model does not include an interaction parameter to describe the salting out of surfactants by ionic solutes, here, the fitting parameters are found from data of the same total composition as the aerosol droplets, which accounts for any salting out. Further development of the Monolayer Model will seek to better describe the salting out of ionic surfactants.

Finally, Fig. 4 compares the surface tension and fractional surface coverage for SDS with 0.5 M NaCl cosolute in a macroscopic solution and in a 10 µm radius droplet using the Simple Kinetic Model. The macroscopic solution surface tension and fractional surface coverage are calculated from the Langmuir isotherm (Eq. 4) using the parameters in Table 1. Fractional surface coverage is defined as $\Theta = \frac{K_{eq}c}{1+K_{eq}c}$. The Simple Kinetic Model, which accounts for the partitioning of surfactant in droplets with high SA-V ratios and agrees well with experimental measurements, is used to calculate the surface tension and fractional surface coverage in a 10 µm radius droplet. In Fig. 4A, depletion of surfactant in the droplet bulk is observed as a shift to higher total surfactant concentrations is required to decrease the surface tension. In this example, the difference in surface tension between a macroscopic solution and 10 µm radius droplet can be as high as 40 mN m$^{-1}$. Figure 4B shows the fractional surface coverage as a function of total surfactant concentration for the macroscopic solution and droplet. As expected from the observed difference in surface tension at total concentrations < 2 mM, more total surfactant is required to cover the same fraction of the surface in a 10 µm radius droplet than in a macroscopic solution.





It is crucial to our understanding of many aerosol processes that, at equilibrium, the number of surface sites
occupied in aerosol droplets is not equal to that of a macroscopic solution. Previous work has shown that the surfactant
partitioning dynamics can reduce the surfactant concentration at droplet surface (Bain et al., 2024b), but bulk depletion can
also reduce the equilibrium surface concentration (Malila and Prisle, 2018). Even at a total concentration equivalent to the
macroscopic CMC, a microscopic droplet containing a strong surfactant is unlikely to have all surface sites occupied. In Fig.
4, for SDS in 0.5 M NaCl, at a total concentration equivalent to the macroscopic CMC, a 10 μm radius droplet is predicted to
have a fractional surface coverage of only 0.16. Assuming the fraction of sites occupied by surface-active molecules at a 10
μm radius droplet interface is approximated by the fraction of occupied sites at the interface of a macroscopic solution would
greatly overestimate the number of occupied surface sites. As droplet radius decreases further (and SA-V increases) these
differences in surface tension and surface coverage widen. Such inaccuracies in surface concentration will affect predictions
of reaction rates for chemical reactions at the droplet interface in addition to predictions of droplet surface tension.

## 4 Conclusion

We experimentally measured the surface tensions of 6 – 10 μm radius aerosol droplets containing ionic surfactants and
cosolutes and compared the results to two independent partitioning models. The macroscopic CMCs of ionic surfactants are
greatly impacted by the presence of salt, which enhances the surface activity of the surfactants. In sea spray aerosol, which
includes both salts and ionic surfactants, interactions between salts and surfactants results in more bulk depletion than for a
droplet of the same size and surfactant concentration but containing an organic cosolute. Bulk surfactant depletion was
observed for all systems except for glutaric acid/SDS in this droplet size range. The total ionic surfactant concentrations
required to reach the minimum surface tension agree with previous observations and predictions for nonionic surfactants,
providing additional evidence that as SA-V ratio increases, the size of the surfactant molecules at the interface plays a larger
role in determining surface coverage than the surface affinity of the surfactant. This phenomenon is due to bulk surfactant
depletion and further highlights that macroscopic measurements are insufficient to predict the equilibrium surface coverage
and surface tension of picolitre volume and smaller droplets containing strong ionic surfactants and salts without applying
partitioning models that account for the SA-V ratio of the droplet.

For aerosol droplets containing ionic surfactants and cosolutes, the Simple Kinetic Model better described the
changing surface tension with total surfactant concentration in the experimental data than the Monolayer Model, likely due
to an incomplete description of salting out in the Monolayer Model or overestimation of the volume taken up by the
surfactant at the interface. However both models predicted similar total surfactant concentrations to reach the effective
droplet CMC. Although $Ca^{2+}$ ions have been shown to affect the partitioning of ionic surfactants in macroscopic solution, at
the low concentrations in aerosol droplets containing the ratio of ions expected from sea spray under high relative humidity
conditions (water activity = 0.99), no difference was observed to the partitioning of SDS or CTAB between NaCl and sea
spray mimic containing droplets. These observations suggest that, at least near the point of cloud droplet activation, the

impact of divalent cations on the surface tension and surface coverage of ionic surfactant-containing droplets is likely small, and NaCl is an acceptable surrogate for sea spray in laboratory studies of surface tension. Surface concentrations of sea spray containing strong, ionic surfactants in dilute droplets can be determined using a partitioning model based on mixtures of NaCl and surfactants rather than sea spray mimic, potentially simplifying required laboratory experiments and modelling

330    efforts aiming to understand surface chemistry at the interfaces of sea spray aerosol.

## Data Availability

All data underlying figures will be uploaded to the University of Bristol Research Data Repository upon acceptance of this manuscript.

## Author Contributions

335    AB, BRB, and NLP acquired funding. AB and BRB designed experiments. AB collected experimental data. AB, KG, KT and NLP developed model code. AB and KG performed modelling. AB prepared the manuscript with contributions from all coauthors.

## Competing Interests

The contact author has declared that none of the authors has any competing interests.

## 340    Acknowledgements

AB acknowledges The Aerosol Society for financial support through a career development grant and the Natural Sciences and Engineering Council of Canada (NSERC) for financial support through a postdoctoral fellowship. AB and BRB acknowledge the European Research Council (ERC) for funding through project AeroSurf (grant agreement ID: 948498). BRB acknowledges the Natural Environment Research Council (NERC) through grant NE/P018459/1. NLP, KG, and KT

345    acknowledge the ERC through project SURFACE (grant agreement ID: 717022) and the Research Council of Finland through grant nos. 308238, 314175, and 335649. NLP and KT further acknowledge the Research Council of Finland through grant no. 316743. The Bristol Aerosols and Colloids Instrument Centre is acknowledged for access to the macroscopic surface tension instrumentation.

350



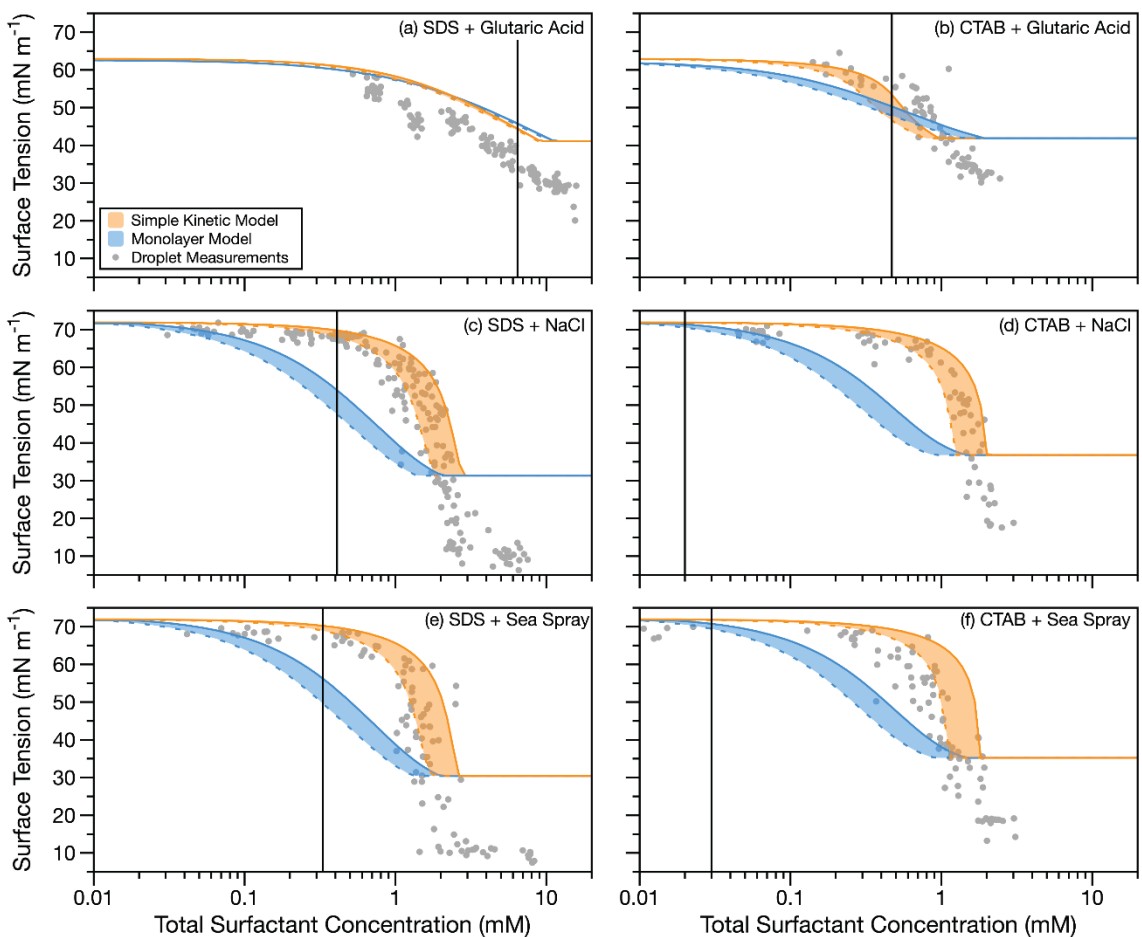

**Figure 1: Comparing Simple Kinetic and Monolayer partitioning models to experimentally measured surface tension for droplets 6 – 10 μm radius. Shaded model areas are predictions for droplets in this size range, with boundaries of 6 μm (solid lines) and 10 μm (dashed lines) radius. Vertical lines indicate the CMC determined from macroscopic measurements. Uncertainties on droplet surface tension measurements are smaller than the data markers.**

355



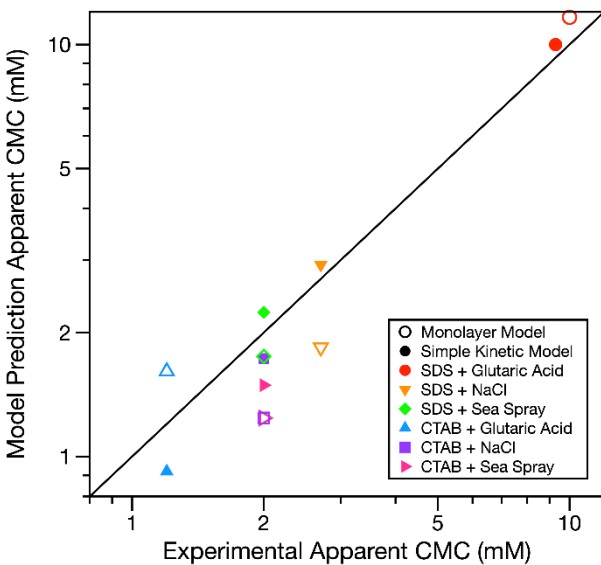

**Figure 2: Comparison of the apparent CMC predictions for 6 µm radius droplets from the Simple Kinetic and Monolayer Models to the experimentally determined apparent CMCs. The black 1:1 line indicates where model predictions and experimental observations are identical.**

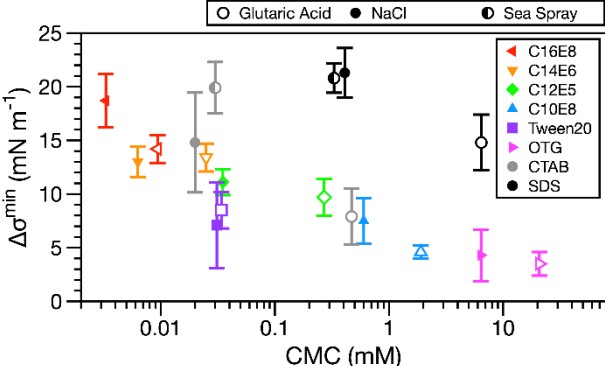

**Figure 3: Difference between minimum surface tension in macroscopic measurements and droplet measurements as a function of the critical micelle concentration for each solution. Ionic surfactant systems investigated in this work are overlaid with the nonionic surfactant data from Bain et al.** (Bain et al., 2023b)**.**





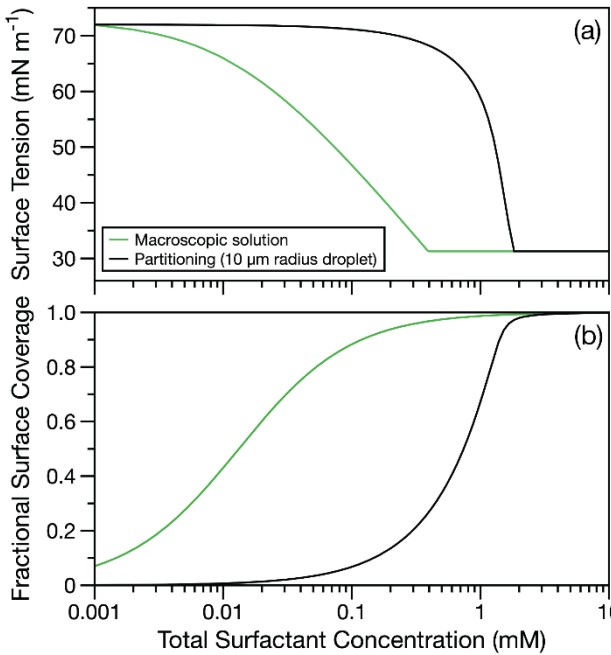

**Figure 4: a) Surface tension and b) fractional surface coverage as a function of total surfactant concentration for SDS with 0.5 M NaCl cosolute in a macroscopic solution from the Langmuir isotherm (green lines) and for a 10 μm radius droplet (black lines). 10 μm radius droplet partitioning is calculated using the Simple Kinetic model.**

**Table 1: Langmuir isotherm parameters, calculated CMCs, and fitting parameters for Monolayer Model.**

|  | Cosolute | $n$ | Bulk CMC (mM) | $\Gamma_{max}$ x 10^6 (mol m^-2) | $K_{eq}$ (m^3 mol^-1) | $b$ (N m^-1) | RMSE (N m^-1) |
|---|---|---|---|---|---|---|---|
| SDS | water | 2 | 9.01 | 5.56 | 0.305279 | -- | 0.00124 |
|  | 0.9 M Glutaric acid | 2 | 6.44 | 2.16 | 0.807898 | 0.0106830 | 0.000747 |
|  | 0.5 M NaCl | 1 | 0.41 | 4.89 | 75.64674 | 0.0121260 | 0.00548 |
|  | 0.48 M Sea spray | 1 | 0.33 | 4.70 | 117.2524 | 0.0116450 | 0.000794 |
| CTAB | water | 2 | 0.96 | 4.70 | 4.129615 | -- | 0.00123 |
|  | 0.9 M Glutaric acid | 2 | 0.47 | 1.01 | 131.5653 | 0.0050263 | 0.00453 |
|  | 0.5 M NaCl | 1 | 0.02 | 3.90 | 1944.691 | 0.0096740 | 0.00154 |
|  | 0.48 M Sea spray | 1 | 0.03 | 2.95 | 5224.200 | 0.0072970 | 0.00148 |





**Table 2: Droplet phase surfactant parameters. Difference between droplet and macroscopic surface tension plateaus ($\Delta\sigma^{min}$) for mixtures with SDS and CTAB. Effective droplet CMC is determined from the intersection of the 6 μm radius droplet bulk depletion prediction and the macroscopic surface tension plateau region curves. Experimental effective droplet CMCs estimated from droplet measurements as the point where the surface tension reaches its minimum value.**

|  |  | $\Delta\sigma^{min}$ (mN m$^{-1}$) | Effective droplet CMC Simple Kinetic (mM) | Effective droplet CMC Monolayer (mM) | Experimental effective droplet CMC (mM) |
|---|---|---|---|---|---|
| SDS | 0.9 M Glutaric acid | 13.0 | 9.27 | 11.65 | 10 |
|  | 0.5 M NaCl | 21.3 | 2.92 | 1.84 | 2.7 |
|  | 0.48 M Sea spray | 20.8 | 2.24 | 1.75 | 2.0 |
| CTAB | 0.9 M Glutaric acid | 7.9 | 0.92 | 1.61 | 1.2 |
|  | 0.5 M NaCl | 14.8 | 1.73 | 1.24 | 2.0 |
|  | 0.48 M Sea spray | 16.9 | 1.49 | 1.24 | 2.0 |


**Table 3: Mean Absolute Error (MAE, mN m$^{-1}$) for model-measurement agreement. Droplet datasets were cut at the effective droplet CMC before calculating MAE.**

|  | SDS |  |  | CTAB |  |  |
|---|---|---|---|---|---|---|
|  | Simple Kinetic MAE | Monolayer MAE | MASE | Simple Kinetic MAE | Monolayer MAE | MASE |
| 0.9 M Glutaric acid | 5.84 | 7.27 | 1.33 | 1.82 | 3.74 | 1.37 |
| 0.5 M NaCl | 2.65 | 8.29 | 3.13 | 1.05 | 6.82 | 6.50 |
| 0.48 M Sea spray | 3.13 | 7.38 | 2.36 | 3.49 | 6.24 | 1.79 |



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
