# Peer review of "Partitioning of ionic surfactants in aerosol droplets containing glutaric acid, sodium chloride, or sea salts"

_EGUsphere, 2024_

## Author Response (AR1)

We thank the reviewers for their feedback and helpful comments on our manuscript. Reviewer comments have been duplicated here in black. Our response to the comment is in blue, and specific changes to the manuscript have been **bolded**.

**RC1**

1. Figure S2. While most of the macroscale parametrizations of surface tension vs SDS concentration yield similar curves that match experimental data, the monolayer model parametrization for CTAB in NaCl and sea salt solution is not good and does accurately fit the experimental data. It is surprising this is not mentioned in the main text and that the 'best fit' parameters are still used to predict droplet surface tension. The authors need to address why the monolayer model does not accurately the macroscale experimental data and discuss how this influences microdroplet surface tension predictions.

    The Langmuir isotherm type parameterization of the data (red lines in Figure S2) is the input for both the Monolayer and the Simple Kinetic Models. The curves for 100 μm radius droplets (blue) and for 100 cm droplets (yellow) are not parameterizations of macroscopic measurements but instead are predictions from the Monolayer Model for particular droplet sizes based on the parameterization shown by the red line. The differences compared to the macroscopic data are due to different predictions about bulk depletion for these systems. In Fig. S2d and f for CTAB in mixtures with NaCl or sea spray, the blue lines (100 μm droplets) are far offset from the macroscopic data because the Monolayer Model predicts strong bulk depletion at this droplet size. When droplet size is increased to 100 cm, the Monolayer Model no longer predicts strong bulk depletion and the yellow line is nearly identical to the parameterization of the macroscopic data (red). We describe these details on page 6 line 166 of the manuscript. To further clarify this point, the caption of Figure S2 has been updated to include the following statement:

    **In panels (d) and (f), strong bulk depletion is predicted by the Monolayer Model for droplet radii of 100 μm, but as the droplet radius is further increased to 100 cm, bulk depletion becomes minimal and the Monolayer Model predictions agree well with the macroscopic data and the Langmuir parameterization.**

2. Line 228. Jacobs et al. (DOI: 10.1021/acs.jpca.4c06210) recently presented a simple expression to predict the effective CMC of surfactants in microdroplets using only the droplet size and macroscale Langmuir parameters. It could be interesting to include a comparison of the experimental effective CMCs to those predicted using the simple expression.

    We have calculated the effective CMC for 6 μm radius droplets and included these results in Table 2. Since the method used by Jacobs et al. (2024) is similar to the Simple Kinetic Model used here, the calculated effective CMCs are generally in close agreement with the effective CMC determined from the Simple Kinetic Model. We do observe an underprediction of the effective CMC using the approach from Jacobs et al. (2024) for the glutaric acid-SDS case by 1.75 mM. The following text has been added to the manuscript (page 9 line 243) to compare these predictions to the results from the Simple Kinetic Model.

**Additionally, we predict the apparent CMCs for 6 µm radius droplets following the approach outlined by Jacobs et al. (2024). In their work, Jacobs et al. (2024) also use a kinetic approach based on the Langmuir isotherm. These predictions are generally in close agreement with the apparent CMCs determined from the Simple Kinetic Model (Table 2).**

3. Figure 2 should include error bars that represent experimental uncertainty (x) and uncertainty in size for the model results (y).

   We have opted to leave out the error bars because many of the points already overlap and including the error bars made the plot very difficult to decipher. We have added the following statement to the caption of Figure 2.

   **Error bars have been omitted for clarity, but uncertainty in the CMC is generally within 10%.**

4. I appreciate that the experimental data were not 'normalized' to the expected minimum surface tension for a surfactant system. However, an obvious experimental artifact/limitation that precludes direct comparison to the two models is the formation of a compressed when droplets collide. Fig. S5 demonstrates that the Langmuir parametrization may extend past the surface excess concentrations that are possible in macroscale solution. Thus, I wonder if it is possible to use this calculate the surface excess concentrations (and by extension surface tensions) in droplets pre-merge geometrically by assuming equal sized droplets and that all molecules at the surface remain at the surface during the measurement.

   The challenge with this approach to determine surface tension of a pre-merged droplet is, experimentally, we do not know that our initial droplets have the same diameter, as we only collect the cavity-enhanced Raman spectrum for the composite droplet. Since we generate our droplets with a medical nebulizer and our two initial droplets are made by the coalescence of droplets from a plume where droplet can vary in size, their sizes are generally similar, but would not be identical. In principle, we could measure the sizes of each initial droplet, but this would be a time-consuming task requiring manual shifting on the portion of the inelastically backscattered light that is delivered to the Raman spectrometer.

   Additionally, while the Simple Kinetic Model appears to follow the trend of the droplet data when surface excess is beyond that expected in a bulk solution, the Langmuir isotherm does not allow for the possibility. The term $\frac{\Gamma}{\Gamma_{max}}$ gets very close to, but does not surpass 1 since it is set equal to $\frac{K_{eq}c}{1+K_{eq}c}$.

5. Line 262. Should read '...with decreasing CMC.'

   This typo has been fixed in the revised manuscript.

6. Consider adding a horizontal line to Figure 4b showing what the fractional coverage is at the CMC.

At the macroscopic CMC, the fractional surface coverage is 0.9998. A line indicating this fractional surface coverage would overlap with the top of the box around the plotting area so we opt not to include one. Instead, we included the following text in the figure caption in the revised manuscript.

**At the CMC, the fractional surface coverage is 0.9998.**

**RC2**

Specific Comments:

1. l.20: The work of (Jacobs, Johnston, et Mahmud 2024) on ionic surfactants is also consistent with your results. You can modify this sentence to take account of this work.

   We have replaced the word nonionic with strong (page 1 line 191).

2. l.69: "atmospheric proxy anionic or cationic surfactants" and l.181: "CTAB and SDS were chosen as commercially available cationic and anionic representatives of atmospheric surfactants." To what extend are they good proxies? For example, in (Gérard et al. 2016; 2019), the critical micelle concentrations (CMC) of the aerosol extracts from filters collected in coastal regions are in the range of a few 100s µM, whereas the CMCs of SDS and CTAB are about 5-10 mM and 1 mM, respectively. Could you comment on that?

   The surfactants in aerosol samples from Lyon, France, reported by Gérard et al. (2019) have CMCs as high as $9.2 \times 10^{-3}$ M (9.2 mM) which is in line with the CMCs of the proxy surfactants used here. Additionally, the many of the surface tension measurements of model primary marine aerosol particles from biologically productive seawater reported by Frossard et al., (2019) do not reach a plateau in the measured surfactant concentration range (up to ~ 0.4 mM), indicating that the CMC, or plateau in surface tension, for the surface active material in this aerosol is greater than 0.4 mM. These proxy surfactants also have O:C and H:C ratios that are in line with surface active organics collected in aerosol samples (Burdette and Frossard, 2021). We have revised the text to further support the selection of these proxy surfactant molecules (page 7 line 192).

   **The O:C and H:C ratios of SDS and CTAB are in the range of surface active organics collected in field measurements (Burdette and Frossard, 2021) and their CMCs are within the range of CMCs reported for PM1 aerosol collected in Lyon, France (Gérard et al., 2019).**

3. It is not clear to me why you write this much about divalent cations. I believe that monovalent cations, and surely anions, also alter the surface properties of surfactants by salting-out effects. According to the Hofmeister series, sodium has a larger salting-out effect than calcium (Hyde et al. 2017). In addition, sulfate, which is also included in the sea salt mimic, has an even stronger salting-out effect than chloride, therefore I wonder why you only mention calcium and not sulfate. Could you clarify this point?

   The surfactants investigated in this work are ionic surfactants which carry a charge (+1 CTAB and -1 for SDS). It is well understood that salts with monovalent ions like NaCl greatly facilitate the salting out of ionic surfactants. (See the works of Nozière et al., 2014; Qazi et al., 2020; Rohde and Sackmann, 1979; Weinheimer et al., 1980, discussed

in the introduction, line 50.) Our interest in divalent cations is due to the bridging interactions that have been observed for $Ca^{2+}$ and anionic surfactants. For example, see the work of Cross and Jayson (1994) and Penfold and Thomas (2022) also discussed in the introduction, line 56).  Such an interaction may be possible for cationic surfactants with $SO_4^{2-}$, however we could not find any literature describing this case. We have revised through the text to include the sulfate ion by replacing 'divalent cations' with 'divalent ions' (e.g. page 2 line 33, page 8 line 198, page 10 line 265 and page 12 line 354).

4. l.89: Where does the $a_w$ = 0.99 comes from?

   In order for droplets to remain stable in the optical trap, we require a cosolute to reduce the water activity.  We have selected a water activity of 0.99 because it is close to saturation, allowing us to probe the surface tension of aerosol approaching activation. The concentration of the cosolute is then chosen to meet this water activity = 0.99 requirement. We have clarified the selection of this water activity in the text (page 3 line 88).

   **These concentrations were chosen to represent water activity near saturation ($a_w$=0.99 for each cosolute), providing information about the surface tension of aerosol droplets near the point of cloud droplet activation.**

5. l.106: "The concentration of surfactant in the droplet was then determined using the molar ratio of surfactant to cosolute in the nebulized solution" In (Bzdek et al. 2020), you justify this assumption on three grounds:

   a. "collection of nebulized aerosol containing glutaric acid and Triton X-100 had the same RI and surface tension as the initial solution and the residual solution in the nebulizer, which is only expected if the glutaric acid:Triton X-100 ratio is conserved upon nebulization"

   b. "if the ratio were not conserved, one might expect substantial variability in retrieved surface tensions among measurements produced from solutions with the same primary solute:surfactant ratio. In fact, the surface tension values of droplets produced by nebulization of the same solution were uniformly consistent (<±3 mN·m−1) over several weeks even using different nebulizers"

   c. "changed. Third, in previous work with mixtures of glutaric acid and NaCl, the retrieved droplet RI and corresponding surface tension gave excellent agreement with model predictions and macroscopic solution measurements, indicating the relative ratio of glutaric acid and NaCl in solution was conserved" upon nebulization

   I must say that I am very surprised that the solute:surfactant ratio remains constant upon nebulization. You have very strong arguments to argue that this ratio is not changed in droplets, yet I would expect aerosol produced by nebulization to be enriched in surfactants, as observed by (Faust et Abbatt 2019). The reason probably lies behind the large volume of droplets you analyze, or comes from your nebulizer. However, I would strongly advise to any future work to check this assumption,

especially for works measuring surface tension of smaller droplets. You checked thoroughly the surface tension of nebulized solution with Triton X-100 for your work in (Bzdek et al. 2020), did you take the same precautions with SDS and CTAB in this work? Could you comment more on your nebulization system?

The experiments performed by Faust and Abbatt use an atomizer to generate their aerosol, whereas we use medical nebulizers. The size distribution of droplets that are generated by these two methods are quite different. Atomizers tend to generate droplets on the order of tens to hundreds of nanometers, while our medical nebulizers generate droplets closer to 4 µm. According to the manufacturer, the mass mean aerodynamic diameter of the aerosol plume generated by our nebulizer is 4.2 µm for a 0.9% saline solution, but the size distribution can vary with chemical composition and ambient conditions.  Organic matter enrichment in sea spray aerosol is generally found to be more prominent in smaller aerosol size fractions (Prather et al., 2013).  We also note that the surfactant to cosolute concentrations ratios are very different in our experiments. Faust and Abbatt worked with equal surfactant: salt ratios while here we work with cosolute concentrations ~50 – 90,000 times greater than the surfactant concentration.

In addition to the control experiments described in (Bzdek et al., 2020), we also compared droplet surface tension measurements for C16E8 containing aerosol droplets generated with nebulizers that generate aerosol using different methods, a mesh grid nebulizer and a sonication based nebulizer (Bain et al., 2023) (Fig. S8). C16E8 is a stronger surfactant than any of the ionic surfactants measured in this work and we would expect any difference in surfactant enrichment due to the nebulization technique to be more prominent than for surfactants with lower propensity for the surface. In that work, we found that both nebulization methods resulted overlapping datasets for the surface tension of droplets (Bain et al., 2023).

If we ever decide to move to smaller droplets generated by a different technique we would perform the necessary control experiments.

The nebulization technique has been described in the previous work cited in the first sentence of Section 2.2 (Bain et al., 2023; Bzdek et al., 2020). We have added the following text to the methods section to further clarify (page 3 line 93).

**Aerosol was generated using a mesh grid medical nebulizer (micro air, Omron).**

6. L.117: You set n=2 for 0.9 M glutaric acid solute, but glutaric acid dissociates into its conjugated base and a proton. At such a high concentration, wouldn't it be preferable to consider this solute as an electrolyte, and set n=1? In addition, setting n=2 leads to lower values for $\Gamma_{max}$ for both SDS and CTAB with glutaric acid compared to other cosolutes (see Table 1). However, if you set n=1, $\Gamma_{max}$=4.32 $10^{-6}$ and 2.02 $10^{-6}$ mol m$^{-2}$ for SDS and CTAB, respectively, which makes more sense to me compared to the values of $\Gamma_{max}$ obtained with other cosolutes.

Glutaric acid is a weak diprotic acid [$H_2A$]. The pKa for the first deprotonation of glutaric acid is 4.35 at room temperature. For a formal concentration of glutaric acid of 0.9 M, the equilibrium concentration of [$H_3O^+$] and [$HA^-$] is 0.00834 M making the equilibrium concentration of  [$H_2A$] = 0.89366, or a fractional dissociation of about 7%. Thus, we expect the glutaric acid cosolute to be largely in the $H_2A$ form which is why we have

chosen to use n=2 for this case. We have revised the text to clarify point (page 5 line 118).

**Here, $n$ = 1 for surfactant mixtures with NaCl or sea spray and $n$ = 2 for surfactant mixtures with glutaric acid. Glutaric acid is a weak diprotic acid with pKa$_1$ = 4.35 at room temperature. For a formal concentration of 0.9 M, we expect only about 7% of the acid to be dissociated.**

7.  On the method used to measure surface tension of droplets. Your results show that the minimum surface tension measured beyond CMC in picolitre droplets is lower than the minimum surface tension reached in macroscopic solutions. In this study, you sum up this observation in the sentence "When the concentration is greater than the effective droplet CMC, nonequilibrium surface concentrations impact the measured surface tension", and call it a bias in the result section (l.239-241). In (Bzdek et al. 2020), you explain that this "may be due to the presence of the surfactant film on the surface of the droplet, which [...] can decrease the droplet oscillation frequency from its true value, resulting in a retrieved surface tension below the true value. Another possible explanation is that the composite droplet surface is slightly enriched in surfactant compared to a fully equilibrated surface. Coalescence produces a composite droplet with a smaller total surface area than the two initial droplets. If surfactant diffusion away from the composite droplet surface is slower than the timescale of the shape oscillation (10 to 100 µs), the composite droplet surface will not reestablish equilibrium, resulting in a tighter packing of surfactant molecules and a reduction in surface tension." Is there any particular reason why the surface of composite droplets would be enriched with surfactants only at concentrations above the CMC, but not below the CMC? At all concentrations, the droplet resulting of the fusion of two smaller droplets has a lower total surface area (by a factor $2^{2/3}$), and a higher surfactant surface content (by a factor 2), therefore a higher surfactant surface concentration (by a factor of $2^{1/3}$), before diffusion of surfactants from the surface to the bulk. In the end, this could lead to a systematic underestimation of droplet surface tension.

We do not see broad evidence for the systematic lowering of surface tension in our droplet measurements over the full total surfactant concentration range measured in this work or in our previous work with nonionic surfactants (Bain et al., 2023). We have used two independent surfactant partitioning models, which only rely on macroscopic surface tension data, not droplet surface tension data, to make surface tension predictions both in previous work (Bain et al., 2024) and in this work. If we consider these 18 surfactant systems, in most cases, we see either good agreement between these models and the experimental data, or the model underpredicts the droplet surface tensions when the droplet bulk concentration is lower than the CMC. If all of the droplet measurements were biased low, we would expect the partitioning models to systematically overpredict the measurements for all surfactants and at all concentrations. Additionally, we point to the recent droplet surface tension measurements presented by Jacobs et al. 2024. These measurements do not rely on the coalescence of two droplets, yet they also find good agreement between their droplet measurements and the predictions of a kinetic partitioning model similar to the one used here. The agreement of a kinetic partitioning model with droplet surface tension

data collected using two different methods provides confidence in our measurements. We have added the following text to highlight this point (page 9 line 254):

**The general agreement between the measured surface tensions of micron sized droplets and the Simple Kinetic Model at surfactant concentration below the effective CMC observed here is further validated by the recent work from Jacobs et al. (2024) who measured the surface tension of SDS and CTAB containing droplets in an electrodynamic balance without the need for a coalescence step.**

8.  l.200 you could also cite the work of (El Haber et al. 2023) in which you can find a lot of complex mixtures of surfactants, salts, and organics.

    We have added the suggested reference.

9.  Figure S4: Could you be more explicit on how you plot this figure? Could you comment on the meaning of ζ? Why did you choose to plot against ζ instead of plotting against the droplet radius?

    ζ is a dimensionless parameter that represents the maximum fractional potential mass lost to the interface and was first introduced by Alvarez et al. (2012). We chose to plot against ζ rather than radius to facilitate comparison with previous work utilizing the simple kinetic partitioning model, where this dimensionless parameter is used (Alvarez et al., 2012; Bain et al., 2024). We have revised the manuscript to clarify the meaning of ζ (page 5 line 134):

    **ζ is a dimensionless parameter that represents the maximum fractional potential mass lost to the interface and $f$ is the ratio of the minimum bulk concentration needed to populate the interface at maximum packing.**

10. l.218: "For SDS with glutaric acid cosolute (Fig. 1A), the macroscopic CMC and effective CMC are in close agreement." It does not seem to me that the droplet surface tension values reach a plateau here. This sentence is a little too affirmative.

    Fig 1 is plotted on a log x-scale so that the full surfactant concentration range can be observed clearly for all mixtures. In panel A, the plateau region is compressed on this scale, but is visible on a linear scale.

[Figure]

To clarify this point we have added the following text to the manuscript (page 8 line 225)

**(note the log scale is required to show the total surfactant concentration range for all systems compresses the plateau region in panel A).**

11. Overall, I think that the end of the article lacks a few sentences about the implications for aerosol-cloud interactions. In light of your results on both non-ionic and ionic surfactants, do you think that surfactants could likely play a role in cloud droplet activation? For example, what would be the concentration of surfactants in the droplet required to reach a 40 mN/m surface tension lowering, with the models and the compounds you studied?

The mixtures of SDS and CTAB with 0.9 M glutaric acid cosolute present in the work have a plateau surface tension above 40 mN/m (Table S2). Instead, we use the Simple Kinetic Model to predict the total surfactant concentration required to reach the minimum surface tension in droplets with wet radius of 0.1 and 1 µm for each mixture. The figure below includes these total surfactant concentrations for the 12 nonionic surfactant systems (Bain et al., 2023) and the 6 ionic surfactant systems in this work (n = 18). The box and whisker plot shows that the Simple Kinetic Model predicts total surfactant concentrations generally in the 10s – 100s of mM range are needed to reach the surface tension of droplets in this size range. These concentrations are in line with previously reported surfactant concentrations from field studies (Gérard et al., 2016, 2019), suggesting the surface tension of ambient aerosol could be lowered by the presence of surfactants during hygroscopic growth. However, in order to determine the impact on cloud droplet activation we must also consider the impact of this partitioning on the total concentration of dissolved solute, which is beyond the scope of this work.

[Figure]

We have added this figure to the manuscript (Fig. 5) and added the following text to the end of the results and discussion section. (page 11 line 316):

**The results presented in this work have focused on droplets in the micron size range, but ambient aerosol particles which act as CCN are typically orders of magnitude smaller. Figure 5 shows predictions of the total surfactant concentration required to reach the minimum surface tension in 100 nm and 1 µm wet radius droplets. Data presented here for SDS and CTAB with glutaric acid, NaCl, and sea spray cosolutes are included, in addition to the 12 nonionic surfactant systems presented by Bain et al. (2023). These surfactants have a wide range of surface activities, with macroscopic CMCs between 1 µM and 10 mM. This box plot shows that total surfactant concentrations in the 10s to low 100s of mM range are required for accumulation mode**

**aerosol droplets to reach their minimum surface tension, regardless of surfactant strength. These concentrations are in line with previously reported surfactant concentrations from field studies (Gérard et al., 2016, 2019), suggesting the surface tension of ambient aerosol could be lowered by the presence of surfactants during hygroscopic growth. However, in order to determine the impact on cloud droplet activation we must also consider the impact of this bulk to surface partitioning on the droplet's water activity, which is beyond the scope of this work**.

Technical corrections

12. l.9: Do you oppose "soluble" to "surface-active"? The term "fully dissolved" would be more appropriate.

    Most organics, even soluble ones like glutaric acid lower the surface tension of water at high concentrations making them both soluble and surface-active. We opt to revise the text to **"The chemically complex organic fraction can contain both soluble and highly surface-active organics"** to better describe the spectrum of surface activity organic molecules can have.

13. l.28: "jet drop aerosol generation pathways have been shown to produce droplets with enriched in surfactant" typo.

    We have removed 'with' from this sentence.

14. l.54-55: "the addition of NaCl to SDS or CTAB solutions [...] enhances the equilibrium and surface concentration", do you mean the surface concentration *at* equilibrium?

    The partitioning kinetics and the equilibrium surface concentration are impacted. We have revised the text to clarify this point (page 2 line 55).

    **affects the partitioning kinetics and enhances the equilibrium surface concentration**

15. l.286: "The Simple Kinetic Model, which accounts for the partitioning of surfactant in droplets with high SA-V ratios and agrees well with experimental measurements, is used to calculate the surface tension and fractional surface coverage in a 10 μm radius droplet" is redundant with the first sentence of the same paragraph.

    We have removed this sentence.

16. l.307: "The macroscopic CMCs of ionic surfactants are greatly impacted by the presence of salt" change "impacted by" by "reduced in".

    We have made this change to the manuscript.

**References Cited**

[revised manuscript text omitted]

---

## Author Response (AR2)

Editor comments have been duplicated here in black. Our response to the comment is in blue, and specific changes to the manuscript have been **bolded**.

**EC**

1. Mean Absolute Scaled Error: to my knowledge, this name is typically used for a slightly different error metric, i.e. when comparing different simulations to some (naive) benchmark. I wonder if it would be better to just speak of, e.g., a MAE ratio here, but please correct me if the MASE is commonly used as the authors use it.

We have replaced every instance of MASE with MAE ratio in the revised manuscript

2. I would appreciate if the names of the nonionic surfactants given in Figs. 3 and 5 would be spelled out somewhere in the manuscript or SI so the reader has this information without consulting Bain et al. (2023). This could be done via their names, structures, or just a description of the compound classes.

We have added the following text to the caption of Fig. 3 to clarify the identity of the nonionic surfactants.

**(CmEn surfactants are linear poly(oxyethylene) alkyl eithers, Tween20 is a commercial surfactant also known as polyoxyethylene (20) sorbitan monolaurate and OTG is octyl-β-D-thioglucopyranoside.)**

3. Side comment on the interplay of Figs. 4 and 5: I would find it instructive if the result of Fig. 5, i.e. minimum surface tension for 1 and 0.1 μm particles, would be represented with a line in Fig. 4, too.

To clarify the interplay between figures 4 and 5 we have added predictions of surface tension and fractional surface coverage for 1.0 and 0.1 μm to Fig. 4. We have revised some of the text in the last paragraph of page 10 and first paragraph on page 11 to discuss these new prediction lines.